# Combined Systemic Drug Treatment with Proton Therapy: Investigations on Patient-Derived Organoids

**DOI:** 10.3390/cancers14153781

**Published:** 2022-08-03

**Authors:** Max Naumann, Tabea Czempiel, Anna Jana Lößner, Kristin Pape, Elke Beyreuther, Steffen Löck, Stephan Drukewitz, Alexander Hennig, Cläre von Neubeck, Barbara Klink, Mechthild Krause, Doreen William, Daniel E. Stange, Rebecca Bütof, Antje Dietrich

**Affiliations:** 1OncoRay—National Center for Radiation Research in Oncology, Faculty of Medicine and University Hospital Carl Gustav Carus, Technische Universität Dresden, Helmholtz-Zentrum Dresden—Rossendorf, 01307 Dresden, Germany; max.naumann@outlook.de (M.N.); elke.beyreuther@uniklinikum-dresden.de (E.B.); steffen.loeck@oncoray.de (S.L.); claere.vonneubeck@uk-essen.de (C.v.N.); mechthild.krause@uniklinikum-dresden.de (M.K.); rebecca.buetof@uniklinikum-dresden.de (R.B.); 2Department of Radiotherapy and Radiation Oncology, Faculty of Medicine and University Hospital Carl Gustav Carus, Technische Universität Dresden, 01307 Dresden, Germany; 3Core Unit for Molecular Tumor Diagnostics (CMTD), National Center for Tumor Diseases (NCT), Partner Site Dresden, 01307 Dresden, Germany; tabea.czempiel@uniklinikum-dresden.de (T.C.); stephan.drukewitz@uniklinikum-dresden.de (S.D.); barbara.klink@lns.etat.lu (B.K.); doreen.william@uniklinikum-dresden.de (D.W.); 4Department of Visceral, Thoracic and Vascular Surgery, Faculty of Medicine and University Hospital Carl Gustav Carus, Technische Universität Dresden, 01307 Dresden, Germany; anna.lamm@uniklinikum-dresden.de (A.J.L.); kristin.pape@uniklinikum-dresden.de (K.P.); a.hennig@biotype.de (A.H.); daniel.stange@uniklinikum-dresden.de (D.E.S.); 5Institute of Radiation Physics, Helmholtz-Zentrum Dresden-Rossendorf, 01328 Dresden, Germany; 6National Center for Tumor Diseases (NCT), Partner Site Dresden, 01307 Dresden, Germany; 7German Cancer Consortium (DKTK), Partner Site Dresden, German Cancer Research Center (DKFZ), 69192 Heidelberg, Germany; 8Institute of Radiooncology—OncoRay, Helmholtz-Zentrum Dresden—Rossendorf, 01307 Dresden, Germany; 9Institute of Human Genetics, University of Leipzig Medical Center, 04103 Leipzig, Germany; 10Clinic for Particle Therapy, University Hospital Essen, Universität Duisburg Essen, 45147 Essen, Germany; 11Department of Genetics, Laboratoire National de Santé, 3555 Dudelange, Luxembourg; 12Institute for Clinical Genetics, University Hospital Carl Gustav Carus, Technische Universität Dresden, ERN-GENTURIS, Hereditary Cancer Syndrome Center Dresden, 01307 Dresden, Germany

**Keywords:** patient-derived organoid, PDAC, pancreatic cancer, radiochemotherapy, 3D cell culture, proton irradiation, translational radiooncology

## Abstract

**Simple Summary:**

Radiotherapy is part of the standard of care for many solid tumors. In pancreatic ductal adenocarcinoma (PDAC), good responses to radiotherapy can only be observed in a minority of patients. In our study, we used PDAC patient-derived organoids (PDO) to investigate alternative radiotherapy approaches for PDAC, such as proton irradiation and combined radiochemotherapy (RCT). Although only very distinct differences in treatment response could be identified, we show the utility of PDOs in translational proton research and found synergistic effects of combined treatments with chemotherapy and proton irradiation in individual PDOs.

**Abstract:**

To optimize neoadjuvant radiochemotherapy of pancreatic ductal adenocarcinoma (PDAC), the value of new irradiation modalities such as proton therapy needs to be investigated in relevant preclinical models. We studied individual treatment responses to RCT using patient-derived PDAC organoids (PDO). Four PDO lines were treated with gemcitabine, 5-fluorouracile (5FU), photon and proton irradiation and combined RCT. Therapy response was subsequently measured via viability assays. In addition, treatment-naive PDOs were characterized via whole exome sequencing and tumorigenicity was investigated in NMRI Foxn1^nu/nu^ mice. We found a mutational pattern containing common mutations associated with PDAC within the PDOs. Although we could unravel potential complications of the viability assay for PDOs in radiobiology, distinct synergistic effects of gemcitabine and 5FU with proton irradiation were observed in two PDO lines that may lead to further mechanistical studies. We could demonstrate that PDOs are a powerful tool for translational proton radiation research.

## 1. Introduction

Pancreatic ductal adenocarcinoma (PDAC) is the seventh leading cause of cancer-related deaths worldwide with almost as many deaths as new cases reported in 2021 [1]. Diagnosis at advanced stages combined with poor treatment outcome makes PDAC one of the most lethal cancer entities today [2]. Radical resection followed by adjuvant chemotherapy (CT) with e.g., gemcitabine or 5-fluorouracile (5FU)/capecitabine is the standard treatment option with some curative potential [3,4,5,6]. However, neoadjuvant CT or radiochemotherapy (RCT) gains importance due to significant benefits in overall survival in resectable and borderline-resectable PDACs [7,8,9].

The anatomy of the pancreas limits the efficacy of radiotherapy (RT) in general [10]. Due to the radiosensitivity of surrounding tissues, it is not possible to apply curative radiation doses to PDAC without severe damage of organs at risk [11,12]. While implementation of modern RT techniques decreased normal tissue toxicity [13,14,15], alternative radiation modalities such as proton therapy show further improvement concerning dose distribution compared to conventional photon irradiation [16,17,18]. A beneficial role of proton therapy in combination with CT was shown for PDAC [19], but there is not enough evidence to finally define the value of RCT in the field of proton therapy [20,21]. In clinical practice, combined RCT is applied in equal treatment schemes regardless of whether photons or protons are used. One reason is the lack of detailed radiobiological knowledge on interactions between drugs and proton irradiation [22]. Systematic studies to understand the relative biological effectiveness (RBE) of protons and the (possible differential) combination effects as well as the development of predictive biomarkers are considered to be among the important avenues in translational proton research [22,23]. For this purpose, the use of meaningful translational models is of utmost importance [20]. Patient-derived organoids (PDOs) are three-dimensional (3D) in vitro tumor models retaining the characteristics of their derived primary tissue [24,25]. In the recent past, organoid biobanks were established to gain further insights in molecular and functional features of PDACs. In addition, new treatment strategies were investigated with organoids to understand individual treatment response within the entity of PDAC [26,27]. By preserving the inter- and intra-tumoral heterogeneity of their tissue of origin they can reflect the individual therapy response of their primary tumor [28,29], including its individual response to RT [30,31]. Moreover, PDOs display cell-cell- and cell-matrix-interactions that are known to influence the radiosensitivity of tumor cells [32,33]. First studies show the potential value of PDOs in translational radiooncology as their response to RT and RCT reflects clinical response parameters of corresponding patients to photon irradiation [34,35,36,37,38].

Although no studies are published yet, PDOs have the potential to be an important translational model to study RCT in proton therapy. Therefore, we investigated the effect of photon and proton RT using PDAC PDOs [27] at the experimental proton beam of the University Proton Therapy Dresden (UPTD) [39]. We show that RCT with protons in combination with clinically relevant chemotherapeutics has supra-additive effects on individual PDAC PDOs. Based on our data, we discuss the potential difficulties analyzing radiation response of PDO cultures and evaluate the value of PDAC PDOs as a translational model in proton research.

## 2. Materials and Methods

### 2.1. Culturing of PDAC Organoids

PDOs were obtained from the PDAC organoid biobank of the Department of Visceral, Thoracic and Vascular Surgery at the University Hospital Carl Gustav Carus of the TU Dresden (VTG) [27,40]. Initial establishment of PDOs from resection material or fine needle aspiration was described in an earlier study [40]. The PDO generation and cultivation were approved by the local ethics committee at Technische Universität Dresden, Germany (#EK451122014). PDOs were embedded in GFR Matrigel (Corning, Corning, NY, USA) and cultured in PDAC culture media based on DMEM/F12 (Gibco, Waltham, MA, USA) supplemented with Pen/Strep (1%, Gibco), GlutaMAX (1%, Gibco), HEPES (1%, Gibco), Wnt3a (100 ng/mL, R&D Systems), R-Spondin (250 ng/mL, PeproTech, Hamburg, Germany), Noggin (100 ng/mL, PeproTech), B27 (1×, Gibco), Nicotinamid (10 mM, Sigma-Aldrich, St. Louis, MO, USA), Gastrin (10 nM, Sigma-Aldrich), N-actyl-L-cysteine (1.25 mM, Sigma-Aldrich), Primocin (100 µg/mL, InvivoGen, San Diego, CA, USA), mEGF (50 ng/mL, Sigma-Aldrich), A-83-01 (0.5 µM, Tocris, Bristol, UK), recombinant human fibroblast growth factor 10 (hFGF-10, 100 ng/mL, PeproTech), murine recombinant epidermal growth factor (mEGF, 50 ng/mL, Sigma-Aldrich) and N2 (1×, Gibco). PDOs were passaged once (DD1522) or twice (DD442, DD314, DD1521) a week as described previously [25] with a split ratio of 1:2.

### 2.2. Organoid Xenograft Tumors and Histology

Animal experiments were approved by the Saxon authorities (Landesdirektion Sachsen, DD24.1-5131/394/73) and are in accordance with institutional, national, and European (EU Directive 2010/63/EU) animal welfare regulations. All NMRI Foxn1^nu/nu^ mice received a whole-body irradiation with 4 Gy two days prior to implantation to suppress the residual immune system. For the subcutaneous implantation of PDOs, confluent wells of a 48 well plate were harvested and implanted in each mouse (Appendix A, *n* = 4 per PDO line). If no tumor growth could be observed, experiments were repeated with a higher number of confluent wells (Appendix A, *n* = 4 per line). The PDOs were harvested and freshly embedded in liquid GFR Matrigel (Corning) on ice. During transplantation, mice were immobilized in a plastic tube to inject 50 µL of the organoid suspension with an insulin syringe on the right hind leg of each mouse. Diameters of the implanted tumors and health condition of mice were monitored twice a week over 122 days for DD314 and DD1521 and 140 days for DD442 and DD1522, respectively. In total, two animals needed to be sacrificed early due to bad general conditions (DD314, DD442). In this case, the small tumor nodules were excised and histologically evaluated for tumor growth.

Tumor material as well as harvested organoids were fixed in formalin and embedded in paraffine. H&E staining was performed according to standard procedures on 3 µm thick tissue slices. Imaging was done with an Axio Imager Z2 (Carl Zeiss, Jena, Deutschland).

### 2.3. Whole Exome Sequencing

DNA was isolated from PDOs and from blood samples of the corresponding patients via DNeasy Blood & Tissue Kit (Qiagen, Hilden, Germany) following the manufacturer’s protocol. The DNA libraries were prepared using the IDT xGen Exome Research Panel v1.0. The whole exome sequencing was performed with the NextSeq 500 Sequencing System from Illumina with a median coverage of 150 reads. Reads were trimmed for quality, aligned to the hg19 human reference genome and further processed using CLC genomics workbench v12.0 (Qiagen). The variants present in the alignment of the blood samples were subtracted from the somatic variants of the PDOs. The retained variants in cancer associated genes were assessed using the criteria of the Guidelines of The American College of Medical Genetics and Genomics [41]. Pathway enrichment analysis was performed using the maftools package v2.6.5 [42,43].

### 2.4. Treatment with RT and RCT

PDO fragments were plated in 96 well plates 24 h prior to treatment. PDO fragment sizes were defined by rinsing PDO fragments through a 50 µm cell strainer before embedding 5000 PDO fragments in 50 µL GFR Matrigel (Corning) per well to standardize the treatment protocol. After reformation of organoids for 24 h, CT with gemcitabine (10 nM) or 5FU (5 µM) was added. The same CT doses were applied for all CT and RCT experiments and for all PDO lines. They were chosen based on dose–effect curves published earlier [40] as a mean dose that decreased viability of the PDO lines to approximately 80% (confirmed with PDO line DD314) under our experimental conditions. PDOs were irradiated 48 h after plating with either photons or protons. Viability was measured using the PrestoBlue assay at day 6, 9 and 13 after treatment start. Media and CT were renewed in all wells after irradiation, on day three after CT and after every viability measurement.

RT with conventional X-rays was performed in plane position with a dose rate of ~1 Gy/min (Maxishot 200Y.TU/320-D03, Yxlon Int. GmbH, Hamburg, Germany; 200 kV, 20 mA; 0.5 mm Cu filter). Proton irradiation was done at the horizontal fixed-beam beamline in the experimental hall of the UPTD. A dedicated double-scattering setup for 150 MeV proton beams [44] was applied to form a laterally homogenous proton field of 10 × 10 cm^2^ with 2.6 cm spread-out Bragg peak (SOBP). Daily quality assessment of the lateral dose distribution of the unaffected irradiation field took place using the Lynx scintillation detector (IBA Dosimetry GmbH, Schwarzenbruck, Germany) at sample position. For irradiation, 96 well plates with organoids were positioned upright in the middle of the SOBP using a range shifter of eleven polycarbonate plates [39]. Daily dosimetry was performed as described earlier [45] and included the cross-calibration of the monitor ionization at beam exit (model 34058, PTW, Freiburg, Germany) to a Markus chamber (model 23343, PTW) at sample position. All organoid irradiations took place at ~10 Gy/min. All reported doses are physical doses without adjustment to RBE. Proton and photon irradiations were performed in independent sessions with individual control groups for every irradiation modality.

For all CT and RCT experiments, the same CT dose was used, which should decrease viability to approx. 80%. The dose was chosen based on the dose–effect curves published earlier [40] that were confirmed with one PDO line (DD314) under our experimental conditions. Medium and CT were renewed in all wells after irradiation, on day three after CT and after every viability measurement.

### 2.5. Viability Assay

Viability of treated PDOs was quantified by measuring NADH dependent metabolic activity via PrestoBlue Cell Viability Reagent (Invitrogen, Waltham, MA, USA). On day 6, 9 and 13 after CT PDOs were incubated with 1:10 PrestoBlue reagent in PDAC culture media for 3 h at 37 °C following the manufacturer’s protocol. Fluorescence was measured at 554/593 nm and images were taken using a Cytation5 (BioTek Instruments, Winooski, VT, USA). Relative viability was calculated after blank subtraction by normalizing every duplicate of treated PDOs to the mean of the duplicate of the untreated control. After every viability measurement, PrestoBlue reagent was replaced with PDAC culture media (including CT in the respective treatment groups) and PDOs were further cultured.

### 2.6. Statistical Analysis

Viability was examined in three independent experiments for each line and each radiation quality. Plotting of dose–response curves and viability after RCT was performed in GraphPad Prism 8.4.0. Single effects of RT and CT were calculated with a two-sided t-test, comparing the treated PDOs with the untreated control. A radiosensitizing or supra-additive effect was defined as an combined effect bigger than the sum of effects after single treatments [46]. To prove statistical significance of combined effects after RCT, generalized estimating equations were used. A linear regression with the measured relative viabilities after RCT as dependent variable and the RT dose, CT and the interaction between RT dose and CT as independent variables was performed. In each case, the independent replicates were considered as repeated measurements. The *p*-value of the interaction term represents the radiation sensitizing effect. An independent working correlation matrix was assumed. All statistical evaluations were carried out with IBM SPSS Statistics (version 28.0.1.1). A *p*-value < 0.05 was considered statistically significant (* *p* < 0.05, ** *p* < 0.01).

## 3. Results

### 3.1. PDOs Mimic Individual Characteristics In Vitro and In Vivo

Four PDO lines (DD314, DD442, DD1521, DD1522) were successfully transferred from the organoid biobank and established in our laboratory. The PDOs grew as sphere shaped cell clusters in vitro, mostly with lumen except for DD1522 (Figure 1). After implantation into NMRI Foxn1^nu/nu^ mice, take rates varied between 0% for DD1522 and 100% for DD1521, thus showing heterogenous tumorigenicity (Table 1 and Appendix A). Xenograft tumors preserved their adenomatous morphology (Figure 1). The proliferation of PDOs in vitro varied between the different lines with DD314 and DD442, showing faster growth compared to DD1521 and DD1522, reflected by lower viability values at day 6, 9 and 13 in culture (Appendix A).

### 3.2. Whole Exome Sequencing Shows Heterogeneous Mutational Patterns

To determine whether the PDAC PDOs show genetic alterations characteristic for PDACs, we performed whole exome sequencing of the PDOs as well as of blood samples of the respective patients to identify somatic mutations. Overall, all PDOs show a molecular profile common for PDACs (Figure 2A). Activating mutations in the oncogene *KRAS* were found in three PDO lines (DD442, DD314, DD1521). They all presented a nucleotide exchange that leads to the commonly known pathogenic substitution at position 12 (p.G12D). DD314 and DD1522 display pathogenic loss of function mutations in *TP53* in the form of a frameshift (DD314) and a known hot spot missense mutation (p.Y220C) (DD1522). The *CDKN2A* gene is affected in two of the organoid samples: The mutation in DD442 leads to a stop gain and the variant found in DD1522 is a deletion of eight bases resulting in a frameshift. In DD442, a likely pathogenic frameshift mutation in *SMAD4* was detected. *ARID1A* is affected by a frameshift mutation in DD1521.

Besides these common gene alterations found in all the examined organoids, they presented further pathogenic variants associated with cancer. In DD442, we found a likely pathogenic mutation in the proto-oncogene *MET*. DD314 showed a hot spot mutation in *U2AF1* (S34F), playing a role in the epithelial–mesenchymal transition and tumor cell invasion [47]. Finally, DD1522 displayed a deletion in the *MAP2K1* gene, which is affected in a high number of cancers and a previously reported hot spot mutation in *GNAS* (R201H) [48]. We further performed pathway enrichment analyses of known oncogenic signalling pathways in the PDAC PDOs and compared the results to the data of the Pancreas Adenocarcinoma cohort of the Cancer Genome Atlas Program (TCGA). As shown in Figure 2B, oncogenic signalling pathway profiles between the TCGA PDAC cohort and the four PDOs are comparable. In both groups, RTK-RAS and TP53 pathways are the most commonly affected pathways. The cell cycle pathway was altered in two of the PDOs. The other investigated pathways were altered in single PDO samples corresponding to a smaller proportion of samples in the TCGA cohort (Figure 2B).

### 3.3. Response of PDAC PDOs to Radiation, Chemotherapy and Combined Radiochemotherapy

Experiments were performed in independent campaigns for the different radiation modalities. In each campaign, radiation was applied in different dose groups and treatment arms with and without CT. Treatments and untreated controls were done in parallel. The schedule is presented in Figure 3A. The complete comprehensive data set is presented in Appendix A for all four PDOs and all conditions.

However, the individual control conditions of each experiment also display the response to single RT and CT treatments, respectively (Figure 3B,C, Table 2): for CT treatment, a heterogeneous response was detected for the PDAC PDOs in our study. The viability of DD314 and DD1521 decreased moderately after treatment with 5FU, whereas gemcitabine treatment of DD1521 caused a very strong decrease in viability. DD442 was not affected by CT at all and did not show any decrease in viability after 5FU or gemcitabine treatment. Interestingly, longer incubation times did not alter the response to CT, although the substances were refreshed with medium exchange. In most cases, the viability loss at day 13 remained at similar levels as day 6 (Figure 3B). RT was performed in different dose groups. By comparing slopes of dose–effect curves, the measured treatment response after RT changed with time. The PDOs showed a heterogenous treatment response on day six after treatment, with DD1521 and DD1522 appearing less affected by RT compared to DD442 and DD314 at this early timepoint. However, on day 13, viability of RT treated organoids did not show large differences between the PDOs independent of the irradiation modality (Figure 3C). When comparing single irradiation doses, DD442 did not show any significant decrease in viability. However, 4 Gy and 6 Gy irradiation decreased the viability of DD314 and DD1522 slightly and independent of the irradiation modality (Table 2).

Response to RCT with the two substances and both beam qualities is shown in Appendix A for three time points after treatment. To evaluate if combined effects are additive or synergistic, day 13 readouts for viability were analyzed for single irradiation dose groups (6 Gy: Figure 4; 4 Gy: Appendix A and Table 3). Due to its very high sensitivity to gemcitabine and a very slow growth in general, which caused large variation of viability measurements in all conditions, DD1521 was excluded from these analyses. In the other lines, the decrease in viability after RCT matched the sum of single effects of RT and CT alone in most cases (Figure 4 and Appendix A). DD314 showed no synergistic effects of either gemcitabine or 5-FU when combined with 6 Gy proton irradiation, respectively (Figure 4). In the photon arm, the single therapies decreased the viability of DD314 already massively, so that no further sensitization could be recognized with this assay. The combination effect is significant (*p* ≤ 0.007, Table 3) which suggests that a combination would lead to an antagonistic effect. However, the results of the 4 Gy group reveal no synergistic effects for DD314 as well (Appendix A and Table 3). DD1522 also showed no sensitization with both irradiation modalities with 6 Gy irradiation. After the combination of 4 Gy protons and 5FU, a synergistic effect was calculated (*p* ≤ 0.002, Table 3). However, the absolute decrease in viability was low (Appendix A) and this combinational effect could not be observed in any other dose groups. However, DD442 reacted differently to the combined treatment. Whereas RCT using proton irradiation with 5FU decreased viability of DD442 only slightly to 80.03% (±28.20%), a massive decrease to 11.92% (±11.19%) with gemcitabine with a significant supra-additive effect (*p* ≤ 0.001, Table 3) was observed. Within the photon experiments, this significant effect could not be seen although a similar tendency was observed (Figure 4). In summary, no clear significant synergism of CT and RT was detected for two out of the three analyzed PDOs. However, DD442 showed remarkable synergistic interactions when treated with gemcitabine in combination with protons.

## 4. Discussion

Neoadjuvant radiochemotherapy [7] and proton irradiation [17,18] are gaining importance as treatment regimens of PDAC. However, the combination of CT with proton irradiation needs a broader understanding to unravel underlying molecular mechanisms and reasons for individual treatment response of patients. Biobanks of PDOs are a promising tool to analyze patient specific response to RCT, including proton irradiation. Besides their ability to act as living avatars for individual patients in the future, organoids are also generally assumed to model tumor response to treatment more realistically than conventional cell culture systems [33]. They presumably mimic the reaction of a tumor to irradiation more authentically due to their 3D growth, cell-cell- and cell-matrix-interactions, making them a potentially valuable tool in translational proton research. In our study, four PDOs could be cultured as described before [28,40], showing their robust transferability to other labs. These four PDOs displayed heterogenous characteristics in terms of mutational landscape. They had common mutations associated with PDAC, e.g., alterations in the five most frequent genes affected by genetic aberrations in PDAC (*KRAS*, *TP53*, *CDKN2A*, *SMAD4* and *ARID1A*) were all present in these PDOs [49]. Furthermore, additional aberrations were detected by whole exome sequencing in comparison to a panel sequencing of selected PDOs, verifying previous results [27].

For three PDAC PDOs, in vivo xenograft growth after subcutaneous implantation in NMRI Foxn1^nu/nu^ mice was achieved. The engrafted tumors showed an adenomatous morphology that was observed earlier in the primary tumors as well [40]. The line that showed no tumor formation (DD1522) should be evaluated in a mouse strain with stricter suppression of the immune system, as the residual immune components in NMRI mice are a potential reason for lower take rates of xenograft transplants [50]. However, commonly used strains with comprehensive immune suppression such as NSG and SCID models depict mutations in DNA damage repair genes, which makes them very sensitive to irradiation [51]. Thus, the three established PDO-derived xenograft models in NMRI form a basis for future translational research in radiation oncology.

Within our study, PDAC PDOs were successfully irradiated with protons and photons as well as treated with RCT. We could show the feasibility of the experimental UPTD setup [39], where organoids are irradiated on a 96 well plate in a vertical position within the middle of a SOPB of an experimental fixed beamline. In a comprehensive experimental setting, the effects of two drugs as single chemotherapy and radiation treatment in comparison to combined RCT using different photon and proton doses in four PDAC PDOs were studied. Although in two lines (DD314, DD1522) no clear significant synergism of CT and RT was visible, DD442 showed striking synergistic interactions when treated with gemcitabine in combination with protons. This indicates a heterogenous effect after RCT in these models, which were already shown to have individual treatment response to CT [40]. We used one single CT dose for our RCT experiments to compare treatment outcome for all PDO lines under the same therapy conditions. Due to the heterogenous responses to CT, some PDOs already show large decreases of viability after single treatment, e.g., DD1521 after gemcitabine. In such cases, identification of further sensitizations after combined treatment is difficult. For future studies, it might be benefical to define individual CT doses for every PDO line when testing for RCT, even though this would limit the comparability between PDOs and make it less feasibile for large-scale testing.

Furthermore, the project was aiming to reveal possible divergent interactions of the beam qualities when combined with chemotherapeutics. Experimental campaigns with photons and protons could not be exhibited in parallel and the limitations of the measured endpoint hinder resolving of RBE-effects from our data. However, organoids may be a potent model system for investigations on RBE. For example, they could be placed at different positions in the Bragg peak and evaluated for induced damage. In this case, the reference irradiations with photons should be done in parallel experiments. In addition, accelerated electrons (LINAC) should be considered as a clinically relevant reference beam modality, as X-rays already differ in RBE from the clinical photon beams as well [52].

Although our data generally underline the feasibility and potential value of PDOs in translational (proton) radiation research, our study reveals limitations of the viability assay for evaluation of therapy response to irradiation. This assay is a standard method in drug testing with PDOs. Indeed, after chemotherapy with 5FU and gemcitabine, all tested models decreased in viability already six days after treatment with no further change afterwards. In contrast, we show time-dependent differences in viability after irradiation. When viability was measured after six days, the PDOs depicted heterogeneous sensitivity. However, taking the heterogenous proliferation rates of PDOs into account, it is obvious that slow proliferating PDO lines (e.g., DD1521, DD1522) revealed higher viability measures compared to fast proliferating PDO lines (e.g., DD442, DD314) at this early timepoint. After 13 days, the viability difference vanished. Whereas the commonly used viability assays can adequately measure early cell death shortly after treatment (the main effect of CT), late cell death after mitotic catastrophe (the main effect of RT) needs an evaluation at a later timepoint. This indicates the necessity of alternative assays for PDOs displaying radiobiological relevant endpoints such as clonogenicity and organoid control. However, such endpoints often rely on the evaluation of (re-)growth of individual specimen [53]. The translation to organoids is challenging, especially for large-scale experiments, as they grow in a matrix and display heterogeneous morphology. To this end, image-based analyses using advanced methods including machine-learning based approaches may have the potential to solve these issues [54].

## 5. Conclusions

In summary, our study shows the feasibility to model proton radiation therapy of pancreatic cancer with patient-derived organoids. Our results indicate the potential value of PDOs to study possible differences in individual treatment response to combined RCT with protons or photons. The observed effects must be further studied regarding their underlying molecular mechanisms, taking their individual mutational pattern into account. For further translation of in vitro findings, we could establish a subcutaneous xenograft model that can be integrated in the UPTD for proton experiments. However, alternative assays to measure radiobiological relevant endpoints of PDO treatment in vitro still need to be established.

## Figures and Tables

**Figure 1 cancers-14-03781-f001:**
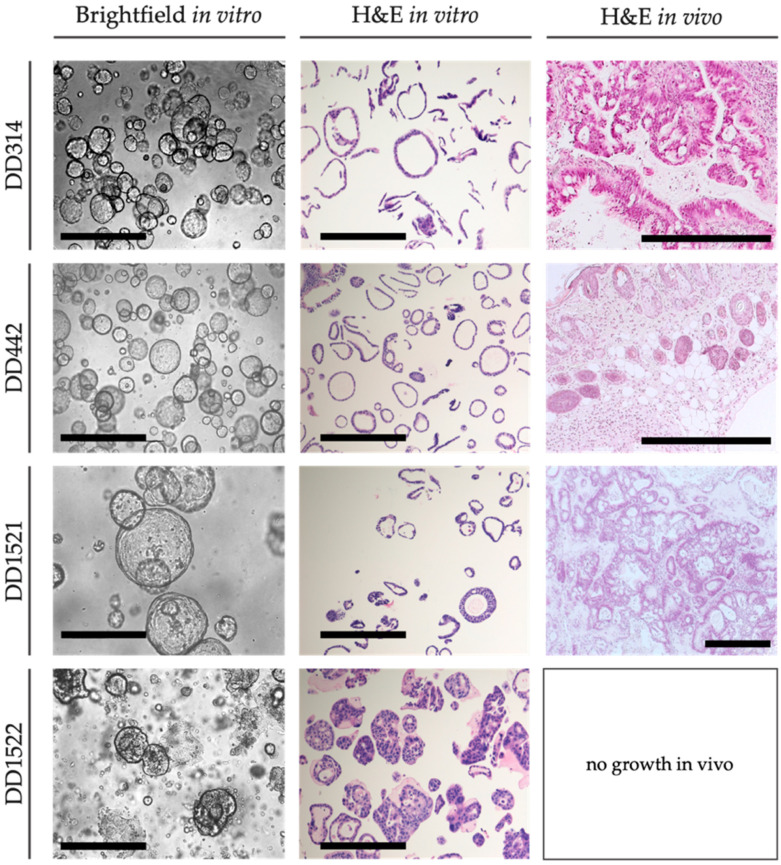
Preservation of PDAC characteristics in vitro and in vivo. Representative brightfield images show the PDO morphology in vitro. Histological sections demonstrate adenomatous morphology in vitro and in vivo (scale bar: 500 µm).

**Figure 2 cancers-14-03781-f002:**
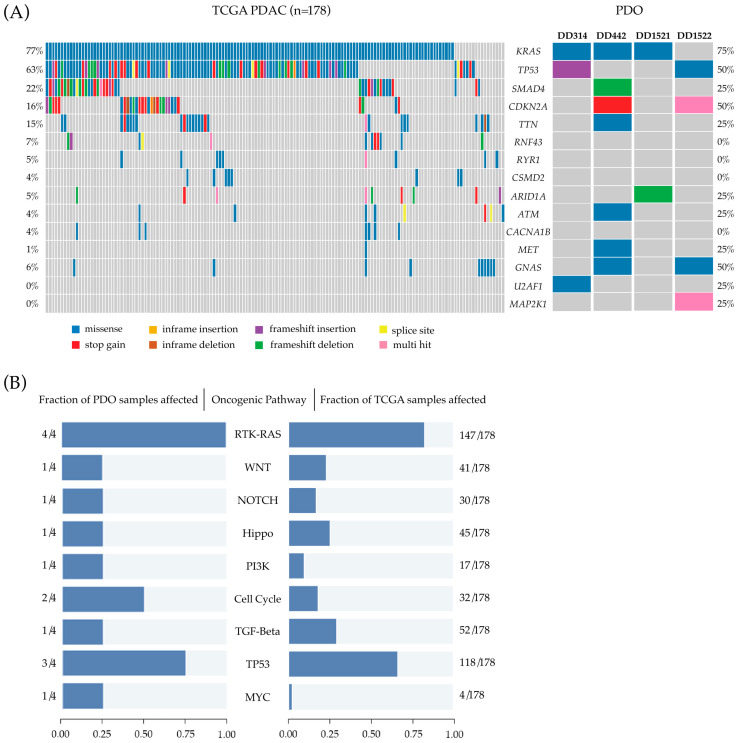
Whole exome sequencing of PDOs (**A**) Oncoplots depicting the 15 most commonly altered genes in The Cancer Genome Atlas (TCGA) cohort of Pancreatic Adenocarcinoma (**left**) compared to the pathogenic and likely pathogenic alterations found by whole exome sequencing in PDAC PDOs, modelling the mutational patterns of their tissue of origin (**right**) (**B**) Visualization of pathway enrichment analysis showing the fraction of affected oncogenic signaling pathways in PDAC PDOs (**left**) compared to the fraction of affected oncogenic signaling pathways in TCGA cohort of Pancreatic Adenocarcinoma (**right**).

**Figure 3 cancers-14-03781-f003:**
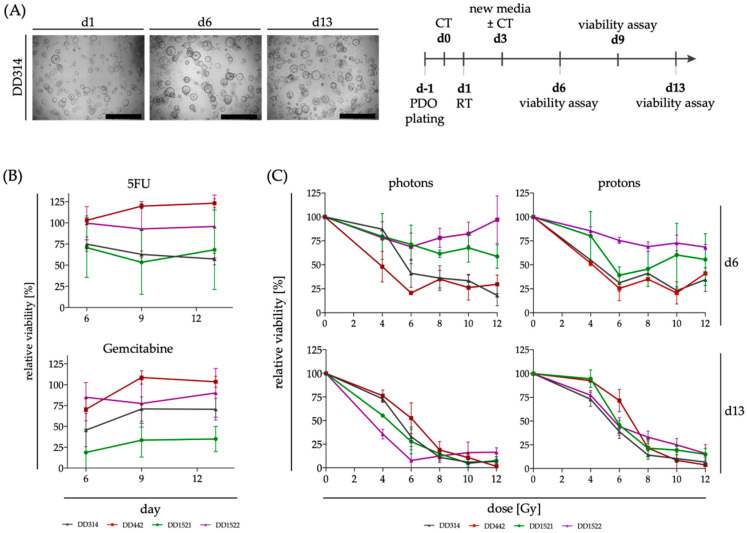
Response of PDOs to single treatments with CT and RT (**A**) PDOs were seeded one day before CT treatment (=d0) and were irradiated after one day. Media was changed after three days and after every viability assay on the days 6, 9 and 13 post CT treatment. Brightfield images show representative untreated PDOs at the different timepoints (scale bar: 500 µm). (**B**) Relative viability ± SD of PDOs after treatment with CT (5FU, 5 µM; gemcitabine 10 nM) at different timepoints after treatment (*n* = 4 for DD1521 and *n* = 5 for the other lines for 5FU; *n* = 6 for DD1522 and *n* = 5 for the other lines for gemcitabine, respectively). (**C**) Dose–response curves of PDOs based on viability measurement with PrestoBlue assay on day 6 and 13 after irradiation with either photons or protons show the dose dependent normalized viability ± SD (*n* = 2 for DD1521 for photons, all other conditions *n* = 3).

**Figure 4 cancers-14-03781-f004:**
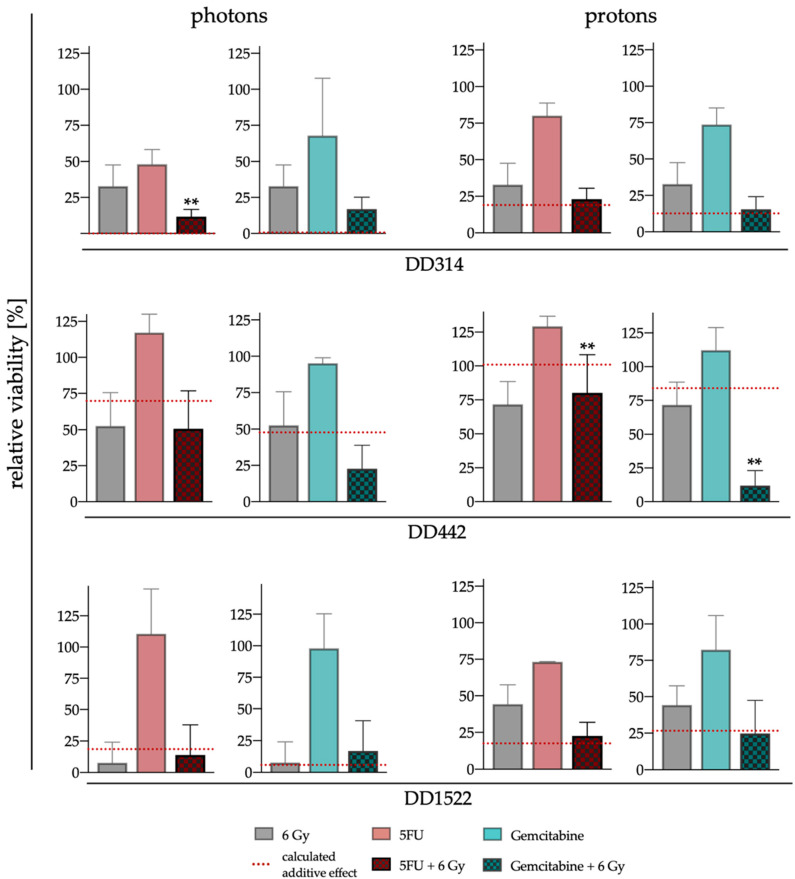
Response of PDOs to RCT with 6 Gy photons or protons. Relative viability was measured on day 13 after RCT with either gemcitabine or 5FU in combination with 6 Gy photon or proton irradiation, respectively ± SD. The red dotted line shows the sum of effects after single treatments (to evaluate if combined effects are synergistic, generalized estimating equation were used, ** *p* < 0.01).

**Table 1 cancers-14-03781-t001:** Subcutaneous xenograft tumor formation in NMRI Foxn1^nu/nu^ mice.

PDO	Number of Implantations	Tumor Bearing Animals
DD314	4	3 *
DD442	4	4 *
DD1521	4	4
DD1522	8	0

* one animal was sacrificed due to bad general condition; tumor growth was histologically confirmed.

**Table 2 cancers-14-03781-t002:** Therapeutic effects after treatment with RT and CT on day 13.

*p*-Value	DD314		DD442		DD1522	
	Photons	Protons	Photons	Protons	Photons	Protons
4 Gy	**0.018**	-	0.063	-	**0.006**	**0.046**
6 Gy	**0.023**	**0.013**	0.100	0.140	**0.015**	**0.027**
gemcitabine	0.372	0.082	0.220	**0.019**	0.928	0.399
5FU	**0.018**	0.082	0.195	0.410	0.714	**0.004**

All evaluated conditions *n* = 3; effects of RT and CT were calculated with a two-sided *t*-test, comparing the treated PDOs with their untreated controls; *p* < 0.05 is marked bold.

**Table 3 cancers-14-03781-t003:** Therapeutic effects after treatment with RCT on day 13.

*p*-Value	DD314		DD442		DD1522	
	Photons	Protons	Photons	Protons	Photons	Protons
gemcitabine + 4 Gy	0.747	-	0.612	-	0.982	**0.046**
gemcitabine + 6 Gy	0.453	0.723	0.135	**<0.001**	0.424	0.710
5FU + 4 Gy	0.586	-	0.416	-	0.209	**0.002**
5FU + 6 Gy	**0.007**	0.519	0.056	**<0.001**	0.823	0.529

All evaluated conditions *n* = 3; generalized estimating equation were used to evaluate if combined effects are synergistic; *p* < 0.05 is marked bold.

## Data Availability

The data presented in this study are available on request from the corresponding author.

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
