# Peer review of "Combined Systemic Drug Treatment with Proton Therapy: Investigations on Patient-Derived Organoids"

_cancers, 2022, doi:10.3390/cancers14153781_

Round 1

Reviewer 1 Report

In this study, the authors successfully constructed four PDO lines (DD314, DD442, DD1521, DD1522) transferred from the organoid biobank and established in our laboratory. Treatment naive PDOs were characterized via whole exome sequencing and tumorigenicity was investigated in NMRI Foxn1nu/nu mice. Four PDO lines were treated with gemcitabine, 5-fluorouracile (5FU), photon and proton irradiation and combined RCT. Therapy response was subsequently measured via viability assays, and distinct synergistic effects of gemcitabine and 5FU with proton irradiation were observed in two PDO lines that may lead to further mechanistical studies. The study demonstrate that PDOs are a powerful tool for translational proton radiation research. However, I still have some concerns.

1.     “Establishment of PDOs from resection material or fine needle aspiration was described earlier”. The establishment of PDOs should be written in a more detailed way.

2.     Why NMRI Foxn1nu/nu mice were used for the animal experiments, but not normal C57BL/C mice.

3.     The result in Fig 2B should be described in a more detailed way in the result section.

4.     Why the response of PDAC PDOs to radiation, chemotherapy and combined radiochemotherapy was not based on the animal experiments.

Reviewer 2 Report

This study developed PDAC patient derived organoids for the investigation of alterantive RT approaches. Although the  use of the organoids system was assessed comprehensively both in vitro and in vivo, I have some questions about the need of this approach.

With experimental details of 4 organoids, the authors have to explain why they focused on proton therapy. When I think of the in vitro setting with a cell plate (the treatment depth is very thin), I wonder if there is sufficient difference between proton and photon energy. Is the underlying radiobiological principle that the authors might initially considered appropriate for the present study design?

In the discussion part, the authors mainly explained the fact that they established the organoids system. Detailed interpretations for Table 2 & 3 are needed.

An additional work for English-editing is needed.

Reviewer 3 Report

The work is interesting and innovative, carried out with multidisciplinary skills.
The article explores a topic that the authors, independently and at different times, even recently, have addressed, thanks to advanced laboratory technologies and innovative projects.
The use of pancreatic ductal adenocarcinoma (PDCA) patient derived organoids (PDO) to compare the relative efficacy of radiotherapy, even in the use of heavy particles, and of chemotherapy alone or integrated with radiotherapy, is a useful argument for future clinical opportunities.
The structure of the article is well organized.
The applied methodology is clearly described.
The bibliography is up-to-date, relevant and appropriately placed.
A rich, punctual and well inserted iconography is a valid support to the arguments dealt with in the text.

Reviewer 4 Report

I should congratulate authors on reasonably well conducted and interpreted experimental work. I have no other comments.

Round 2

Reviewer 1 Report

The authors have addressed the comments. There is no further comment.

Reviewer 2 Report

No other comments are needed for this revised version of manuscript.